# Individual and community-level factors associated with animal source food consumption among children aged 6-23 months in Ethiopia: Multilevel mixed effects logistic regression model

**Hassen Ali Hamza**[1], **Abdu Oumer**[2]*, **Robel Hussen Kabthymer**[3], **Yeshimebet Ali**[4], **Abbas Ahmed Mohammed**[5], **Mohammed Feyisso Shaka**[3], **Kenzudin Assefa**[2]

1 Quality Improvement Unit Coordinator at Mekane-selam General Hospital, Mekane-selam, Ethiopia,
2 Department of Public Health, College of Medicine and Health Sciences, Wolkite University, Southwest, Ethiopia, 3 School of Public Health, College of Medicine and Health Sciences, Dilla University, Dilla, Ethiopia,
4 School of Public Health, College of Medicine and Health Sciences, Wollo University, Dessie, Ethiopia,
5 Department of Midwifery, College of Medicine and Health Sciences, Dilla University, Dilla, Ethiopia

* phnabu2@gmail.com

**Data Availability Statement:** The dataset generated and/or analyzed during the current study

## Abstract

### Background

Diversified diet in childhood has irreplaceable role for optimal growth. However, multi-level factors related to low animal source food consumption among children were poorly understood in Ethiopia, where such evidences are needed for decision making.

### Objectives

To investigate the magnitude and individual- and community-level predictors of animal source food (ASF) consumption among children aged 6–23 months in Ethiopia.

### Methods

We utilized a cross-sectional pooled data from 2016/19 Ethiopia Demographic and Health Surveys. A stratified two-stage cluster design was employed to select households with survey weights were applied to account for complex sample design. We fitted mixed-effects logit regression models on 4,423 children nested within 645 clusters. The fixed effect models were fitted and expressed as adjusted odds ratio with their 95% confidence intervals and measures of variation were explained by intra-class correlation coefficients, median odds ratio and proportional change in variance. The deviance information criterion and Akaike information Criterion were used as model fitness criteria.

### Result

in Ethiopia, only 22.7% (20.5%-23.9%) of children aged 6–23 months consumed ASF. Younger children aged 6–8 months (AOR = 3.1; 95%CI: 2.4–4.1), home delivered children

is accessed from http://dhsprogram.com/data/available-datasets.cfm on a reasonable request.

**Funding:** The authors received no specific funding for this work. The funders had no role in study design, data collection and analysis, decision to publish, or preparation of the manuscript.

**Competing interests:** The authors declare that they have no competing interest.

**Abbreviations:** AIC, Akaike's information criterion; ANC, Antenatal care; AOR, Adjusted odds ratio; ASF, Animal Sources Food; CI, Confidence interval; COR, Crude odds ratio; DIC, Deviance information criterion; DHS, Demographic and Health Survey; EA, Enumeration areas; EDHS, Ethiopia Demographic and Health Survey; ICC, Intra-class correlation coefficient; MOR, Median odds ratio; PCV, Proportional change in variance; PSU, Primary sampling unit; SE, Standard error; UNICEF, United Nations Children's Fund; VIF, Variance inflation factor; VPC, Variance partition coefficient; WHO, World Health Organization.

(AOR = 1.8; 1.4–2.3), from low socioeconomic class (AOR = 2.43; 1.7–3.5); low educational level of mothers (AOR = 1.9; 95%CI: 1.48–2.45) and children from multiple risk pregnancy were significant predictors of low animal source consumption at individual level. While children from high community poverty level (AOR = 1.53; 1.2–1.95); rural residence (AOR = 2.2; 95%CI: 1.7–2.8) and pastoralist areas (AOR = 5.4; 3.4–8.5) significantly predict animal source food consumption at community level. About 38% of the variation of ASF consumption is explained by the combined predictors at the individual and community-level while 17.8% of the variation is attributed to differences between clusters.

## Conclusions

This study illustrates that the current ASF consumption among children is poor and a multiple interacting individual- and community level factors determine ASF consumption. In designing and implementing nutritional interventions addressing diversified diet consumption shall give a due consideration and account for these potential predictors of ASF consumption.

## Background

Under-nutrition remains a significant public health concern in Ethiopia among children [1], with 36.8%, 21.1%, and 7.2% of children are victims of stunting, underweight and wasting by 2019 and its adverse consequences respectively [2]. Major burden of malnutrition starts during pregnancy and early childhood where diversified and nutritious dietary habit could promote optimal growth, cognitive development and economic productivity [3].

The first 1000 days of early life is a critical period where the impacts of malnutrition will be devastating and proceeds to adulthood [4, 5]. Among these age groups, infant should be fed exclusively for the first six months and complementary feeding with a diversified diet is recommended starting from six months. Also, the infancy and the early childhood period is a risky period for widespread stunting and wasting due to problems in child feeding [6, 7]. Undiversified food consumption is one of the major predisposing factors for children aged 6 to 23 months of age [8].

Diversified diet in early childhood has irreplaceable role for optimal growth. Animal source foods (ASFs) have a high amount of critical nutrients that are more bio-available than nutrients from plants source foods [9]. ASFs are better sources for high quality proteins and essential amino acids [10], better and optimal child growth. In addition, ASFs can significantly improve nutritional status especially in children [11], due to the fact that it provides a high concentration of indispensable macro and micronutrients, to meet their daily requirements [12, 13].

ASFs (now onwards refers to consumption of egg and/or other flesh foods) have been linked to improved nutritional status in children, including reduced stunting [9, 14]. Prevalence of stunting is significantly lower in children who had consumed ASFs in the last 24 hours as compared to those who had not [10]. A review study indicated that better ASF consumption is linked to a positive physically observable anthropometric advantages and improved cognitive function among undernourished children from low-income settings [13]. A review on intervention trial of ASF supplement for children showed to improvement in the length, weight and Height for age Z scores [15].

Despite a better ASF production in the country, the rapidly rising food prices and poor economical access to diversified ASF consumption makes ASF consumption low, where the diet of many rural households were limited to maize and legumes [16]. Taking the burden of under-nutrition in early childhood and the limited diversified dietary habit and consumption into consideration, consumption of ASF could significantly contribute to the overall quality of diet and the long-term strategies to control malnutrition in the country.

However, the practice of ASF consumption is a function of various factors which might have a potential to hinder consumptions. So far, limited attempts were tried to understand the problem with the use of small-scale surveys [17–19], despite the multifaceted nature of the issue, where individual or household level factors only may not be sufficient. Hence, analysis approach considering individual and community level factors and design could allow to better understand the drivers of ASF consumption. In low-income settings like Ethiopia cultural factors like decision making power of women's, religious factors like fasting and socio-economic factors like wealth status, ownership of livestock and other factors like low level of nutrition knowledge were among the reported factors known to affect ASF consumption [20, 21].

Hence, these factors could easily be understood if we could analyze using multi-layered approach like community/cluster level, household level and individual level factors [12, 22] at national scale where this gear towards informed decisions. Besides, the result may be used to design strategies and formulate programs that might improve the level of ASF consumption in young children. Previous study by Pott et.al indicated that limited individual level factors (religion, residence, child age) were associated with a better ASF consumption [20]. In addition, Workicho et.al identified residence, livestock ownership, socioeconomic class and literacy positively predicted better ASF consumption [16]. However, these studies were limited in some regions and does not involve a more robust assessment of community level factors, which may determine the ASF consumption due to the culture, livestock availability or other potential factors [23]. Furthermore, the effect of factors like high-risk fertility behaviors and multi-level factors on ASF consumption of children was poorly investigated, where studies showed that 57% to 72% of women had high risky fertility behavior [24, 25], which may predispose to a widespread food insecurity and malnutrition among children [24] Thus, this study aims to determine the level of ASF consumption and factors affecting it among children aged 6–23 months in Ethiopia.

## Methods

### Data source, study design and sample points

This study utilized a cross-sectional pooled data from Ethiopia Demographic and Health Surveys (EDHS) conducted in 2016 and 2019. The data were extracted from www.measuredhs.com.The data were nationally-representative population-based household surveys. The standard DHS have large sample size (usually between 5,000 and 30,000 households) carried out about every 5 years [26]. A community-based cross-sectional study design was used. The DHS surveys are based on a stratified two-stage cluster sampling design, where independent multi-stage samples are selected per strata. Within each stratum implicit stratification is applied to make sure that the selected primary sampling units are representative of different geographic levels and areas. Stratified primary sampling units (clusters) were sampled in the first stage and households in the second stage [27]. This two-stage sampling points allows to have a representative sample with a reduced sampling errors and appropriate coverage for target population. Sample size for this complex survey with clustering estimated with design effect (Deft). To prevent bias, no replacements or changes of the preselected households were allocated in the implementing stages [27]. The EDHS surveys used sample weights to account for complex

survey design, survey non-response, and post-stratification for representativeness of the samples. The study population for this study were youngest living child age 6–23 month who is living with the mother (KR file) 24 hours preceding the interview. After data cleaning and exploration, a total weighted sample of 4,423 children aged 6–23 months were included in the survey and in our analysis.

## Measures of variables

**Dependent variable.** According to WHO and UNICEF, ASF consumption among children age 6–23 months is defined as the percentage of children 6–23 months of age who consumed egg and/or flesh food on the previous day. This indicator is based on consumption of food groups 5 (flesh foods) and 6 (eggs) described in indicator 8 on minimum dietary diversity [28]. Children are counted as "consumed ASF" if either food group has been consumed, otherwise children are counted as or "not consumed ASF" [29, 30]. Hence, the dependent variable (outcome variable) is dichotomized as ("0"—consumed ASFs, "1"—do not consume ASFs).

**Independent variables.** Based on reviewed literature, both individual and community-level predictor variables were considered in our analysis. From the individual-level variables, child's factor (age of child, sex of child, previous birth interval, and birth order), maternal factors (high-risk fertility behaviors, maternal age at birth) [24, 25, 31], socioeconomic factors (wealth index, maternal education, maternal occupation, exposure to media), and health service factors (place of delivery, and antenatal visit). We adopted the concept of high-risk fertility behaviors from DHS surveys, which considers three parameters, mother's age at birth, birth order, and birth interval, to define high-risk fertility behaviors. The high-risk fertility behaviors were categorized as: no extra risk, unavoidable risk, single high-risk and multiple high-risk. The presence of any of the following 4 parameters was considered as a single high-risk fertility behavior: mother's age <18 only, mother's age >34 only, birth interval <24 months only and birth order above three. The combinations of two or more risk parameters are referred to as multiple high-risk fertility behaviors [16, 31].

Community-level variables (community poverty level, community-level education) were created from individual-level variables by aggregating them at the cluster (community) level by using the bysort command. We obtained the proportion of each community-level characteristic and the values were ranked into tertiles as low, medium, and high. Community-level education, which was the proportion of women with secondary or higher education in the community and categorized into tertiles as low, medium, or high. Similarly, the community poverty level was categorized into tertiles and classified as low, medium, or high poverty level [32].

*Data management and analysis.* The unit of analysis for this study was children aged 6–23 months in pooled DHS data and the data was exported and analyzed using Stata/SE version 14.0. Sample weights were applied for descriptive statistics adjusting for non-proportional allocation of the sample and non-response rate in all analyses. This makes sample data representative of the entire population. Categorization was done for continuous variables and further re-categorization was done for categorical variables. Descriptive analysis was carried out to present the data in frequencies and percentages.

Since EDHS data is nested data (4,423 children nested within 645 clusters) and a two-level clustered dataset with a multistage sampling design, we applied multilevel modelling, which acknowledges the nesting with in the survey. In nested data (hierarchical data), analyzing variables from different levels at one single common level with 'standard' analysis method is inadequate, and leads to loss of statistical power and conceptual problems (ecological fallacy and atomistic fallacy) [33, 34]. Thus, we fitted a two-level multilevel mixed-effects logistic

regression (random-intercept model), with the log of the probability of inadequate ASF consumption was modeled using a two-level multilevel model as follows [34]:

$Y_{ij} = \beta_0 + \beta_1 x_{1ij} + \mu_{oj} + e_{oij}$ where, $Y_{ij}$ is our outcome variable (animal source food consumption): the animal source food consumption for a child aged 6–23 months living in cluster j, $\beta_0$ is the intercept, $\beta_1$ is the coefficient of explanatory variable $x_1$, the part of the equation involving the β-coefficients, $\beta_0 + \beta_1 x_{1ij}$, is called the fixed part of the model because the coefficients are the same for everybody; the residuals at the different levels, $\mu_{oj} + e_{oij}$, are collectively termed the random part of the model. We fitted four models for a mixed effects modeling for nested data to determine the model that best fits the data. Null model (M0) or the intercept-only model: a model with no explanatory variables, model-I: a model with only individual-level factors, Model-II: a model with only community-level factors, and model-III: a combined model that control the effects of both individual and community-level predictor variables on ASF consumption among children aged 6–23 months. The Stata command–"meqrlogit" was used to fit these models. The results of fixed effects were expressed as adjusted odds ratio (AOR) with their 95% confidence intervals (CIs).

The measures of variation were expressed as Intra-class Correlation Coefficients (ICC) or Variance Partition Coefficients (VPC), Median Odds Ratio (MOR), and Proportional Change in Variance (PCV). The ICC and VPC can be computed for random intercept models. In the case of a random intercept model, in a logistic regression model with no predictors, the ICC or VPC equals: $VPC = ICC = \frac{level-2\ residual\ variance}{level-2\ residual\ variance+level-1\ residual\ variance}$, for logit model, the level-1 residual variance is $\frac{\pi^2}{3} = \frac{(3.14)^2}{3} = 3.29$. Therefore, $ICC = \frac{level-2\ residual\ variance}{level-2\ residual\ variance+3.29}$ [33, 34]. The MOR is a measure of heterogeneity while the VPC (ICC) is a measure of components of variance (clustering) that considers both between- and within-cluster variance. The MOR depends directly on the area level variance (the variance of the highest-level errors) and can be computed with the following equation: $MOR = \exp(\sqrt{(2 \times Vc)} \times 0.6745] \approx \exp(0.95\sqrt{(Vc)})$ where, $Vc$ is the between cluster variance. The proportional change in variance (PCV) is the percentage of proportional change in variance of subsequent models with respect to the empty model. The PCV can be computed by the equation: $PCV = \frac{(VA-VB)}{VA} \times 100$, where VA = variance of the initial model (empty model), and VB = variance of the model with more terms (consecutive models) [35]. The deviance Information Criterion (DIC), log likelihood and Akaike Information Criterion (AIC) were used to select the best model that explained the variation in ASF consumption well. The model with the smallest AIC is chosen as the one which fits the best. Models with a lower deviance fit better than models with a higher deviance [36, 37]. Variance Inflation Factors (VIF), Standard Error (SE), and Variance Correlation Estimator (VCE) were estimated to assess risk of multicollinearity among predictor variables.

**Ethical considerations.** Permission to access and download the EDHS datasets was obtained from DHS program. The accessed data were used for the purpose of registered research paper only. Confidentiality of the data were kept and no effort was made to identify any household or individual respondent interviewed in the survey. The data were not passed on to other researchers without the written consent of DHS. The data were fully accessed from www.dhsprogram.com with the respect to the data sharing policy.

## Results

### Socio-demographic characteristics of participants (children with their mothers)

A total of 4,423 children aged 6–23 months nested within 645 clusters (primary sampling units) from 2016 and 2019 Ethiopia DHS pooled data were included in our analysis. Out of the

total participants, 2,154(48.7%) were male children. About 816(18.4%) children were in the age group 6–8 months,720 (16.3%) in the age group 9–11 months, 1,636 (37%) in the age group 12–17 months, and 1,251 (28.3%) were aged 18–23 months. The majority, 3,972 (89.8%) of mothers attended up to primary education while 3,649 (82.5%) were from rural areas.

### Health related characteristics

From total participants, only 1,682(38%) of mothers had attended at least four ANC visits during their recent pregnancy. Concerning the place of delivery, nearly more than half of the mothers, 2,453 (55%) gave birth at home, whereas, 43% of households were in the lowest wealth quintile. A total of 1,608 (36.4%) falling into birth with any single high-risk category and 910 (20.6%) falling into births with any multiple risk category. The highest proportion, about 1,814 (41%) and 1,229 (28%) of children were from a community where the poverty level is high and low respectively (Table 1).

### National animal source food consumption pattern among children

In this study, a total of 22.7% (95% CI: 20.5%-23.9%) of infant and young children age 6–23 months consumed ASF (either food groups: food groups 5 (flesh foods) or 6 (eggs) has been consumed) based on the standards in Ethiopia. An estimated 8.7% and 17.5% of children consumed flesh foods and egg products during the previous day of the interview, respectively (Table 2). The percentage of children with adequately ASF consumption is highest among children from the richest household family (37.2%) and lowest from the poorest household family (14.5%). The result also showed that ASF consumption improves with increasing household wealth status (Fig 1). A large-scale regional variation (41.7% highest in Addis Ababa and 5.9% lowest in Somali) was observed in ASF consumption among children aged 6–23 months in Ethiopia (Fig 2).

### Individual-and community-level predictors of ASF consumption

**Measures of association (fixed effects) from multilevel logistic models.** Bivariable multilevel mixed-effect logistic regression was computed and factors with a lower p-value (p-value below 0.25) and/or variables with strong theoretical relation with ASF consumption [38], and previously identified predictor variables were used as a cutoff to fit multivariable multilevel mixed-effects logistic regression to control confounding effects. Consequently, age of the child (in months), having four ANC visit, place of delivery, household wealth index, maternal education level, access to all three media at least once a week, place of residence, community poverty level, community-level education, double risk (birth order 4+ & age 34+), double risk (spacing <24 & order 4+), births with any multiple risk category, any avoidable risk category, and region were eligible for multivariable analysis. We fitted four consecutive multilevel mixed-effects logistic regression models. The intercept-only model (null model: _M0) (without predictor variables), model-I (a model with only individual-level factors), model-II (a model with only community level factors) and model-III (a combined model that controls the effects of both individual- and community-level predictor variables) on ASF consumption were fitted. Ultimately, associations with a p-value below 0.05 were used to identify a statistically significant predictors of ASF consumption among children aged 6–23 months in the full model (parsimonious model).

In our analysis, age of the child (in months), at least four ANC visits, place of delivery, the household wealth index, maternal education level, community poverty level, births with any multiple risk category, access to all three media at least once a week, place of residence, high-risk fertility behaviors, and the region were statistically significant predictors of ASF

**Table 1. Individual-and community-level characteristics of participants, pooled data from Ethiopia DHS 2016 & 2019 (n = 4,423).**

| Variables | Weighted frequency (%) |
|---|---|
| **Sex of child** | |
| Male | 2,154 (48.7%) |
| Female | 2,268 (51.3%) |
| **Age of child in months** | |
| 6–8 months | 816 (18.4%) |
| 9–11 months | 720 (16.3%) |
| 12–17 months | 1,636 (37%) |
| 18–23 months | 1,251 (28.3%) |
| **Mother's age at birth** | |
| > = 18 years | 4,243 (96%) |
| <18 years | 179 (4%) |
| **Maternal educational level** | |
| Primary education or less | 3,972 (89.8%) |
| Secondary education and above | 450 (10.2%) |
| **Place of residence** | |
| Urban | 774 (17.5%) |
| Rural | 3,649 (82.5%) |
| **Attended 4[+] ANC visit** | |
| No | 2,740 (62%) |
| Yes | 1,682 (38%) |
| **Place of delivery** | |
| Health facility | 1,855 (42%) |
| Home | 2,453 (55%) |
| Other | 115 (3%) |
| **Household wealth index** | |
| Poorest | 979 (22%) |
| Poorer | 937 (21%) |
| Middle | 925 (21%) |
| Rich | 797 (18%) |
| Richest | 783 (18%) |
| **High-risk fertility behaviors risk category** | |
| No extra risk | 1,135 (26%) |
| Unavoidable first birth risk | 770 (17.4%) |
| Any single high-risk category | 1,608 (36.4%) |
| Any multiple risk category | 910 (20.6%) |
| **Accesses to all three media at least once a week** | |
| No | 4,395 (99.4%) |
| Yes | 27 (0.6%) |
| **Community poverty level** | |
| High | 1,814 (41%) |
| Moderate | 1,380 (31%) |
| Low | 1,229 (28%) |
| **Community-level education** | |
| Low | 4,180 (94.5%) |
| High | 242 (5.5%) |

**Table 2. ASF consumption among children aged 6–23 months in Ethiopia: Based on pooled data from 2016 and 2019 Ethiopia Demographic and Health Survey (n = 4,423).**

| Diets | Frequency (n) | Percentage (%) |
|---|---|---|
| Animal Source Food Consumption | | |
| No | 3,421 | 77.3% |
| Yes | 1,002 | 22.7% |
| Food Groups 5 (flesh foods) Consumption | | |
| No | 4,036 | 91.3% |
| Yes | 386 | 8.7% |
| Food Group 6 (eggs) Consumption | | |
| No | 3,647 | 82.5% |
| Yes | 775 | 17.5% |

consumption among children aged 6–23 months in the final model (full model). Our finding revealed that, age of the child had significantly associated with ASF consumption among children aged 6–23 months. Younger children aged 6–8 months (AOR 95% CI: 3.1(2.4, 4.1) and 9–11 months (AOR = 1.54; 95% CI: 1.20–1.97) had a 3.1- and 1.5-times increased risk of inadequate ASF consumption as compared to older children aged (18–23 months).

This study also showed that, children of mothers who attended primary educational had 1.9 times the risk of inadequate ASF (AOR = 1.90; 95% CI: 1.48–2.45) as compared to children whose mothers attended at least secondary education level. Those children whose mothers gave birth at home had 80% more risk of having inadequate ASF consumption (AOR = 1.80; 95% CI: 1.4–2.3). Additionally, the household wealth index was significantly associated with ASF consumption. Accordingly, the odds of having inadequate ASF consumption among children living in the poorest households were 2.4 times (AOR = 2.43; 95% CI: 1.70–3.50) more likely than children from the richest households. Moreover, the odds of having inadequate ASF consumption among children aged 6–23 months living in the poorer households had 70% (AOR = 1.70; 95% CI: 1.22–2.37) more risk as compared to children living in the richest household. Furthermore, those children whose mothers had less than four ANC visits during pregnancy were 1.4 times more likely to have inadequate ASF consumption (AOR = 1.40; 95% CI:

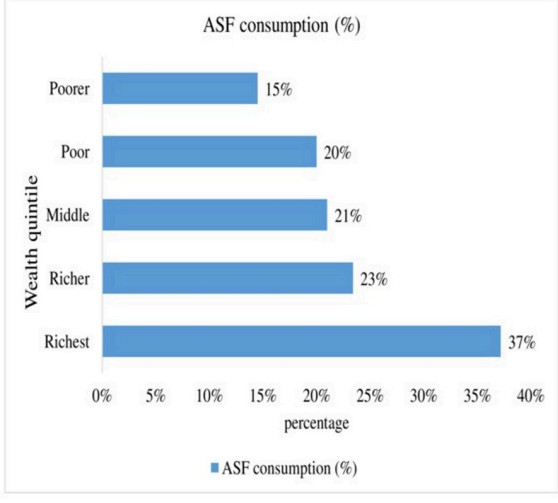

**Fig 1. ASF consumption by household wealth status among children aged 6–23 months in Ethiopia.**

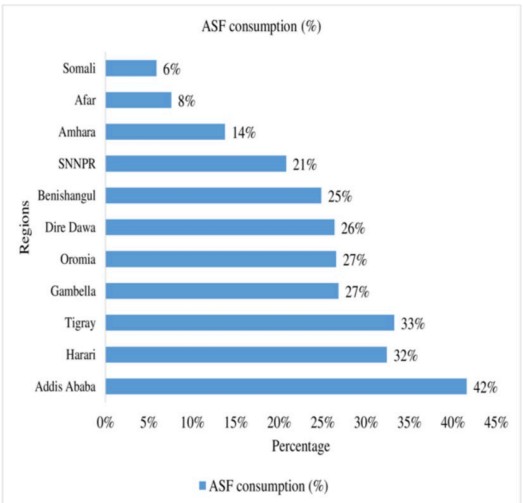

**Fig 2. Interregional variation in ASF consumption among children aged 6–23 months in Ethiopia.**

1.02–1.83) as compared to those whose mothers had at least four ANC visits during the last pregnancy. Likewise, the region was significantly associated with ASF consumption. Also, the odds of being inadequate for ASF consumption among children aged 6–23 months whose mothers had accesses to all three media at least once a week was 72% less likely to have inadequate ASF consumption than (AOR = 0.28; 95% CI: 0.12–0.65) compared to whose mothers had access none of the three media. Likewise, community poverty level was significantly associated with ASF consumption among children aged 6–23 months in Ethiopia. Accordingly, the odds of having inadequate ASF consumption among children aged 6–23 months living in high community poverty level (cluster) were 53% (AOR = 1.53; 95% CI: 1.20–1.95) more likely than children from low community poverty level. This result also revealed that, children who lived in rural areas were 2.2 times (AOR = 2.20; 95% CI: 1.70–2.80) more likely to have inadequate ASF consumption compared with children from urban areas. Similarly, children born from multiple risk category, double risk with birth order 4+ & age above 34, and double risk with spacing <24 months, order 4+ were significantly associated with ASF consumption among children 6–23 months in Ethiopia (Table 3).

**Random effects (measures of variations) and model fitness.** We fitted the mixed-effects models for binary responses with- meqrlogit- Stata command. Mixed-effects logistic regression is logistic regression containing both fixed effects and random effects. We found the likelihood-ratio test (LR test) statistically significant (p-value < 0.05), with better fitted regression model better than the traditional. Based on the latent variable estimation method, the constant quantity $\pi^2/3 \sim 3.29$ for the lower-level variance. Thus, for a two-level multilevel logistic regression model with a random intercept, ICC = VPC = Vc/ [vc+3.29]. The ICC (VPC) quantifies the degree of homogeneity of the outcome within clusters. From the intercept-only model (null model), we found the variance partition coefficient (VPC): 17.8%. This implied that 17.8% of the variation in animal source food consumption among children aged 6–23 months to be attributable to differences between clusters (primary sampling units). In our data, the estimate of the between clusters variance was 0.71(0.51, 0.98: at p-value <0.001) for empty model (M0). The corresponding MOR was 2.23, indicating the odds of inadequate ASF consumption was 2.23 times higher in the children with the higher propensity the outcome of interest compared to those children with lower propensity. The MOR is a measure of heterogeneity in both between- and within-cluster variance. In this study, the final model (M3) revealed

**Table 3.** Multilevel mixed-effects logistic regression modeling of individual- and community-level factors associated with no ASF consumption among children aged 6–23 months in Ethiopia, based on pooled data from Ethiopia DHS 2016 & 2019 (n = 4,423).

| Individual-and community-level characteristics | COR [95% CI] | Full model: AOR [95% CI] |
|---|---|---|
| Age of child (in months) | | |
| 6–8 | 3.1 (2.36,4) | 3.1 (2.4, 4.1) |
| 9–11 | 1.46 (1.15,1.87) | 1.54 (1.2,1.97) |
| 12–17 | | 1.23 (1.02,1.5) |
| 18–23 | 1.25 (1.03,1.51) 1 | 1 |
| 4+ ANC visit | | |
| No | 1.98 (1.67, 2.33) | 1.22 (1.01,1.5) |
| Yes | 1 | 1 |
| Place of delivery | | |
| Health facility | 1 | 1 |
| Home | 2.5 (2.1, 2.96) | 1.8 (1.4,2.3) |
| Household wealth index | | |
| Poorest | 5.04 (3.93,6.46) | 2.43 (1.7, 3.5) |
| Poorer | 2.43 (1.88,3.14) | 1.7 (1.22, 2.37) |
| Middle | 2.1 (1.61,2.71) | 1.44 (1.03,2.0) |
| Richer | 1.79 (1.37,2.33) | 1.37 (1.01, 1.87) |
| Richest | 1 | 1 |
| Maternal education level | | |
| Primary & below | 3.3 (2.67, 4.06) | 1.9 (1.48, 2.45) |
| Secondary+ | 1 | 1 |
| Accesses to all three media at least once a week | | |
| No | 1 | 1 |
| Yes | 0.19 (0.08,0.43) | 0.28 (0.12, 0.65) |
| Place of residence | | |
| Urban | 1 | 1 |
| Rural | 2.75 (2.24,3.37) | 2.2 (1.7,2.8) |
| Community poverty level | | |
| Low | 1 | 1 |
| Moderate | 1.6 (1.2, 2.03) | 1.3 (1.01,1.69) |
| High | 2.27 (1.8, 2.9) | 1.53 (1.2, 1.95) |
| Community-level education | | |
| Low | 1 | 1 |
| High | 2.4 (1.8, 3.4) | 1.1(0.8,1.5) |
| Birth with any multiple high-risk fertility behaviors category | | |
| Double risk, birth order 4+ & age 34+ | | |
| No | 1 | 1 |
| Yes | 1.45(1.08,1.94) | 1.4 (1.02,1.83) |
| Double risk, spacing <24, order 4+ | | |
| No | 1 | 1 |
| Yes | 2.5 (1.7,3.7) | 1.93 (1.3, 2.9) |
| Births with any multiple risk category | | |
| No | 1 | 1 |
| Yes | 1.8 (1.4, 2.2) | 1.4 (1.02,1.83) |
| Region | | |
| Tigray | 1 | 1 |
| Afar | 8.5 (4.98,14.38) | 8.2 (4.8,13.9) |

(*Continued*)

**Table 3.** (Continued)

| Individual-and community-level characteristics | COR [95% CI] | Full model: AOR [95% CI] |
|---|---|---|
| Amhara | 3.7 (2.4,5.6) | 3.3 (2.2,5.0) |
| Oromia | 1.23 (0.87,1.74) | 1.1 (0.8,1.5) |
| Somali | 5.74 (3.66,9.0) | 5.4 (3.4,8.5) |
| Benishangul | 1.6 (1.05,2.32) | 1.4 (0.9,2.0) |
| SNNPR | 1.79 (1.24,2.56) | 1.6 (1.1,2.3) |
| Gambela | 1.11 (0.74,1.66) | 1.2 (0.8,1.8) |
| Harari | 0.74 (0.5,1.1) | 0.9 (0.6,1.3) |
| Addis Ababa | 0.59 (0.4, 0.9) | 1.2 (0.8,1.8) |
| Dire Dawa | 1.34 (0.87,2.05) | 1.8 (1.14,2.70) |

that a higher proportional change in variance (PCV) with 38%, showing about 38% of the variation of ASF consumption explained by the combined predictors at the individual-and community-level. For the reason that the models are nested, we used the –2 log likelihood (DIC) based statistic to compare models with regard to goodness of fit. Since the DIC in final model (M3) was the lowest in our analysis, the final model (M3) better fit to the data (Table 4).

## Discussion

The current analysis indicated that the ASF consumption of children in Ethiopia is 22.7%. This showed that nearly one in five children had ASF consumption. Diversified diet consumption is a proxy indicator for micronutrient adequacy, including iron, zinc and others which promotes optimal linear growth and development [18]. Low ASF consumption is more prevalent in more food insecure regions of the country (Afar, Somali, and other regions). In fact, ASFs are a significant contributor for the daily intake of bioavailable nutrients, with the potential to address the existing burden of stunting (37%) and protein-energy malnutrition (7.2%) in the country [1]. ASF consumption has the potential to decrease the stunting as compared to those with low ASF consumption [18], which emanates from eating monotonous plant source diets which in turn usually threatens nutrient bioavailability.

**Table 4. Random effects (measures of variations) and model fitness for no ASF consumption among children at primary sampling units (clusters) level by a multilevel mixed-effects logistic regression modeling based on pooled data from Ethiopia DHS 2016 & 2019.**

| Parameters | Null model (empty model) | Model-III |
|---|---|---|
|  | model-0 | (Final model) |
| **Random-effects (measures of variations)** | | |
| Cluster level variance (SE) | 0.71 (0.12) | 0.44 (0.1) |
| PCV (%) | Reference | 38% |
| ICC or VPC (%) | 17.75% | 11.8% |
| MOR | 2.23 | 1.88 |
| **Fit criteria (model diagnostics)** | | |
| Loglikelihood (LL) | -2202.076 | -1980.142 |
| DIC (-2LL) | 4,404.152 | 3,960.284 |
| AIC | 4408.153 | 4018.284 |

**AIC:** Akaike's information criterion, **DIC:** Deviance information criterion, **ICC:** Intra-class correlation coefficient, **MOR:** Median odds ratio, **PCV:** Proportional change in variance, **SE:** Standard error, **VPC:** Variance partition coefficient

Other study also showed that ASF consumption is low while the ASF expenditure is raising. Dairy products (14.7%), beef (3.2 kg), and egg (0.3%) were the most commonly consumed foods [39]. A better ASF consumption of 64% and 18% among Urban and rural was reported among school age children [40].

Despite a wide variation in ASF consumption by wealth status and residence, low ASF consumption is highly prevalent in different parts of the country. In this study, we have observed a very low ASF consumption in pastoralist communities like Somali region as compared to other parts of the country. Apart from the resource scarcity related factors, one of the reasons for this disparity could be measurement of ASF, as we do not considered milk and milk products, but it is widely used in pastoralist communities of Ethiopia [41]. Another study also indicated that children from the highest wealth quintile were more likely to have a better expenditure for ASF (16.5% and 4.9% of household expenditure) [39]. It also indicated that food support schemes like the productive safety net improve the dietary diversity and consumption of ASF, targeting parts of the society having low income [16].

In Ethiopia, the price index of ASF is rising at alarming rate with an average raise in price of ASF by 36% [17]. Strategies to increase access to ASF targeting the poor are crucial. Long-term strategies like employment schemes, and food for work approaches can aid in increasing diversified diet consumption in the short-term. However, long-term strategies shall be in place along with the short-term approaches in order to expect a lasting result [42, 43]. Furthermore, improving quality livestock production can increase access to ASF and consumption can alleviate malnutrition [44]. Such approaches allows sufficient intake at affordable price for better nutritional outcome and health status of children [41, 45].

Studies have shown that livestock and asset ownership can potentially improves ASF consumption [18]. Sustainable strategies through diversified agricultural production coupled with enhanced behavioral change communication approaches can improve ASF availability and consumption [46]. In addition, the concept of nutrition sensitive agriculture will play a mounting role if implemented with sectoral collaboration. However, the affordability of high quality, nutritious foods is threatened by the worsening market prices of food items in the country. The policy directions should encompass a multipronged approach to stabilize and increases the affordability of animal source foods.

The results of our analysis in the final model (model III) showed that both individual-level factors (age of the child, ANC visits, place of delivery, household wealth index, maternal education level, access to all three media at least once a week, and high-risk fertility behaviors), and community-level factors (community poverty level, place of residence, and region) were significant predictors of ASF consumption. This study showed that the proportional change in variance (PCV) of the full model was responsible for about 38% of the log odds of low ASF consumption in the communities. The outcome of MOR, a measure of unexplained cluster heterogeneity, is 2.23 and 1.88 in the null model and final model respectively. Thus, the results of MOR in this analysis indicate that there is unexplained variation between the clusters of the community. The VPC results also showed a significant cluster-level variance, which is a minimum precondition for a multilevel modeling.

In this study, children aged 6–8 months and 9–11 months had a 3.1- and 1.5-times increased risk of low ASF consumption as compared older children aged (18–23 months). This showed that consumption of ASF increases with age. Similar findings were found in studies conducted in Ethiopia. The possible justification could be lack of nutritional knowledge [19, 20, 22, 47, 48], late initiation of nutrient-dense complementary foods [49] and social norms and beliefs [20, 21]. The other possible reason for this association may be due to food taboos around ASF consumption in children [48]. Thus, Behavioral Change Communication (BCC) interventions are needed to improve the caregivers' nutritional knowledge as effective

strategy for infant and young children feeding practices. The finding of this study revealed that children whose mothers had at least four ANC visit and gave birth at health facility had a protective effect against low ASF consumption. Nutritional counseling and better exposure to nutrition information may have a crucial role for better ASF consumption among children. In addition, those families with access to ANC service are more likely educated, and from high socioeconomic class which may have a better ASF access than others [50]. The possible pathways (mechanisms) of this association may be due to ease of access to health information that may improve the likelihood of mothers practice of proper child feeding [49].

Similarly, this study revealed that children from poorest and poorer households were more likely to have low ASF consumption compared with children from richest households. This finding was similar with studies done in Cambodia, Ethiopia, and Indonesia [9, 20, 21, 49]. As the household wealth status increases, consumption of ASF increases. The possible mechanisms might be due to the agricultural policy and ASF prices and attitude related issues like use of ASFs as a source of income rather than feeding their own children [20, 21, 51]. Because the imbalance of the price and demands of ASFs in the community are high, almost every product of livestock is commonly intended for market purposes. As the result, the consumption of ASFs in poor households is hampered among children [20, 21]. The other possible justifications may be cultural factors like low women's empowerment, leading to decreased prioritization of household nutrition[48, 51]. In contrary, when, women have greater decision-making power over household expenditures and control over resources, it is likely that the households would allocate larger proportions of their income and resources to improve their children's diets [51]. This study found that, children of mothers who attended at least secondary educational level and have accesses to all three media at least once a week had less likely to have low ASF consumption as compared with their counter parts respectively. These findings were similar to that of studies conducted in Nepal [52]. These might be due to the fact that educated mothers might have better nutritional knowledge about the nutritional advantages of ASF consumption [20, 21, 51]. In addition, the role of media in raising awareness and shaping the practice of mothers towards better child feeding is substantial.

In Ethiopia, 35% [53], 57% and 36.8% [2] are a victims of Zinc deficiency, anemia and stunting, respectively where the positive role of ASF consumption in alleviating these problem is undisputable. Also, with this currently very low ASF consumption and less diversified foods, the 2030 targets for Zero hunger and stunting will be ideal without sustainable approaches to improve the dietary intake of children. Promoting nutrition sensitive agriculture, rearing livestock for own consumption; can improve consumption and income for better livelihoods. Above all, improving the livestock productivity, especially targeting the pastoralists through improved technologies and approaches allows widespread availability of ASF in the country [51]. Such strategies could sustainably improve the livestock productivity, and empowers women to address malnutrition in the developing countries [54].

Despite, the availability of livestock products, the household level understanding on the necessity for ASF, and household heads preference for market instead of feeding their child is the major factor in the majority of the rural area [52]. However, it is believed to be related to the deep routed problems of food insecurity, lack of knowledge and poverty which plays a great role for low level of diversified food consumption in developing country. Strengthened poverty reduction schemes and behavioral change communication (BCC) to improve the practice of proper child feeding should be conjoined [55]. In addition, consumption of ASF among children is low and have a widespread spatial variation. There should be a targeted strategies to increase access to ASF, and increase ASF production. Some cultural perspectives might also hinder access to ASF in some areas of the country. There should be a strengthened nutritional counseling and nutrition education strategies for an improved access and consumption of ASF

for children. Furthermore, strong regulatory frameworks shall be in place to stabilize the increasing food prices in the country.

## Acknowledgments

We received the data from the Demographic and Health Surveys (DHS) Program. We would like to thank the DHS for granting us with the data.

## Author Contributions

**Conceptualization:** Hassen Ali Hamza, Abdu Oumer, Robel Hussen Kabthymer, Yeshimebet Ali, Abbas Ahmed Mohammed, Mohammed Feyisso Shaka, Kenzudin Assefa.

**Data curation:** Hassen Ali Hamza, Abdu Oumer, Robel Hussen Kabthymer, Yeshimebet Ali, Abbas Ahmed Mohammed, Mohammed Feyisso Shaka, Kenzudin Assefa.

**Formal analysis:** Hassen Ali Hamza, Abdu Oumer, Robel Hussen Kabthymer, Yeshimebet Ali, Abbas Ahmed Mohammed, Mohammed Feyisso Shaka, Kenzudin Assefa.

**Funding acquisition:** Hassen Ali Hamza, Abbas Ahmed Mohammed, Mohammed Feyisso Shaka.

**Investigation:** Hassen Ali Hamza, Abdu Oumer, Robel Hussen Kabthymer, Yeshimebet Ali, Abbas Ahmed Mohammed, Mohammed Feyisso Shaka, Kenzudin Assefa.

**Methodology:** Hassen Ali Hamza, Abdu Oumer, Robel Hussen Kabthymer, Yeshimebet Ali, Abbas Ahmed Mohammed, Mohammed Feyisso Shaka, Kenzudin Assefa.

**Project administration:** Hassen Ali Hamza, Robel Hussen Kabthymer, Yeshimebet Ali.

**Resources:** Hassen Ali Hamza, Abdu Oumer, Robel Hussen Kabthymer.

**Software:** Hassen Ali Hamza, Abdu Oumer, Robel Hussen Kabthymer, Yeshimebet Ali, Abbas Ahmed Mohammed, Mohammed Feyisso Shaka, Kenzudin Assefa.

**Supervision:** Hassen Ali Hamza, Robel Hussen Kabthymer, Yeshimebet Ali, Mohammed Feyisso Shaka, Kenzudin Assefa.

**Validation:** Hassen Ali Hamza, Abdu Oumer, Robel Hussen Kabthymer, Yeshimebet Ali, Abbas Ahmed Mohammed, Mohammed Feyisso Shaka, Kenzudin Assefa.

**Visualization:** Hassen Ali Hamza, Abdu Oumer, Robel Hussen Kabthymer, Yeshimebet Ali, Abbas Ahmed Mohammed, Mohammed Feyisso Shaka, Kenzudin Assefa.

**Writing – original draft:** Hassen Ali Hamza, Abdu Oumer, Robel Hussen Kabthymer, Yeshimebet Ali, Abbas Ahmed Mohammed, Mohammed Feyisso Shaka, Kenzudin Assefa.

**Writing – review & editing:** Hassen Ali Hamza, Abdu Oumer, Robel Hussen Kabthymer, Yeshimebet Ali, Abbas Ahmed Mohammed, Mohammed Feyisso Shaka, Kenzudin Assefa.

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
