## [Decision Letter · Decision Letter 0]

22 Nov 2021

PONE-D-21-25620Individual and community-level factors associated with animal source food consumption among children aged 6-23 months in Ethiopia: Multilevel mixed effects logistic regression modelPLOS ONE

Dear Dr. Abdu,

Thank you for submitting your manuscript to PLOS ONE. After careful consideration, we feel that it has merit but does not fully meet PLOS ONE’s publication criteria as it currently stands. Therefore, we invite you to submit a revised version of the manuscript that addresses the points raised during the review process.

We look forward to receiving your revised manuscript.

Kind regards,

Sukumar Vellakkal

Academic Editor

PLOS ONE

“I have read the journal's policy and the authors of this manuscript have the following competing interests. The authors declare that they have no competing interest. All authors agreed on the submission of the manuscript.”

Reviewers' comments:

Reviewer's Responses to Questions

**Comments to the Author**

1. Is the manuscript technically sound, and do the data support the conclusions?

Reviewer #1: Partly

Reviewer #2: Yes

2. Has the statistical analysis been performed appropriately and rigorously? 

Reviewer #1: No

Reviewer #2: Yes

3. Have the authors made all data underlying the findings in their manuscript fully available?

Reviewer #1: No

Reviewer #2: No

4. Is the manuscript presented in an intelligible fashion and written in standard English?

Reviewer #1: No

Reviewer #2: Yes

5. Review Comments to the Author

Reviewer #1: It is great to see more research in nutrition using MLM methods. The discussion is rich in detail and the overall paper contributes importantly to growing interesting in ASF.

Above all, I would suggest a thorough copy editing from someone knowledgeable in the field. It would be made much stronger with some language revisions.

I have two, relatively large, suggestions for revision:

1. I would be interested to see the analysis re-done including dairy products. Dairy is also an ASF and an important one at that, and its exclusion as part of ASF is confusing.

2. It would be very helpful to see your iterations on model fit. Many of the variables seem to overlap or don't have a strong theoretical grounding, and some (e.g. livestock ownership) seem to be missing.

I've provided detailed feedback in the attached comment.

Reviewer #2: This is a very good piece of work. I have enjoyed reading it. The topic has important policy implications, the method suits the nature of the data structure, and the effort put into the manuscript to make a robust explanation is admirable. My general feedback is very good with mainly one editorial/additional estimation correction required apart from the routine editorial details I found. For this main comment, please see 2.1. Other than this, the followings are some of the comments I put forward in each subheading:

1. General comments

1.1. “Animal source food consumption is very low” (line 47) – reference? Compared to?

1.2. “despite a better livestock population, diversified diet consumption is poor..” should it be diversified ASF? ( line 76)

1.3. Which time period are the data points referring? (line 78)

1.4. “So far, limited attempts were tried to understand the problem with the use of small-scale surveys, despite the multifaceted nature of the issue, where individual or household level factors only may not be sufficient” (line 86 – 88). What are these studies? A few citation of these studies will be ideal. How is your study unique from these studies?

1.5. “Nationwide study addressing the factors that affect the level of ASF consumption has not been conducted in Ethiopia” (line 97 – 98) I am afraid I can’t agree to this claim. For example, Workicho et.al (2016) Household dietary diversity and Animal Source Food consumption in Ethiopia: evidence from the 2011 Welfare Monitoring Survey, Potts et.al (2019) Animal Source Food Consumption in Young Children from Four Regions of Ethiopia: Association with Religion, Livelihood, and Participation in the Productive Safety Net Program are a few examples. You can cite these researchers and point out what makes your study unique.

1.6. “Currently, the price of ASF is rising alarmingly in the country above 36%”. Lin 381. This should be re-written. There is no single ASF price as such unless it is a price index, I think.

1.7. “In Ethiopia, 35% [37], 57% and 36.8% [9] are a victims of Zinc deficiency, anemia and stunting…”line 438, who are these figures referring to? Of the Ethiopian population, Ethiopian children or ?

2. Methodology

Measurement of the dependent variable – while a dichotomized variable is a good starting point, it is also crude as it aggregates measurement and somehow undermines heterogeneity, for example in the type of ASF consumed here. In this regard, as a robustness extension,

2.1. In addition, “A large scale regional variation (..) in Addis Ababa and … lowest in Somali was observed in ASF consumption aged 6-23 months.” (line 262 – 264). And “Low ASF consumption is more prevalent in more food insecure regions of the country (Afar, Somali, and other regions).” (Line 361 – 362). “Apart from the resource scarcity related factors, one of the reasons for this disparity could be measurement of ASF, as we do not considered milk and milk products widely used in pastoralist communities of Ethiopia”.

This provides a wrong information and introduces systematic bias. Afar and Somali constitute one of the largest livestock population making children more likely to consume milk and milk products. These regions do not have similar wealth of chicken and are less likely to consume egg.

As a result, either

• Bring the definition of ASF used in this manuscript forward explicitly at the start (Mainly in the dependent variable section) and give a concerted direction to your reader what you mean by an ASF or

• Include milk and milk products and do additional analysis,

• Leave Afarand Somali from your analysis. My advice for you is to include milk and milk products and re do the analysis while keeping the current results as they are and see how things change. This will give you percentage values of ASF consumption and regression estimates to which you can compare results.

2.2. Will a separate regression (as a dichotomous variable) be an extension of this study or be used as a robustness check? For example, estimating the same model for the consumption of Egg and other flesh foods alone.

2.3. Counting the number of ASF and generating an ordered variable to run multi-level ordered logistic regression provide similar results?

2.4. “Only 27 (0.6%) of mothers had accesses to all three media at least once a week” which three media? I think this needs re-phrasing the statement. Looking at Table 1, this variable does not show variability worth to have it in a description. 99% said No to media access. This requires re categorization or will not make a good predictor. This is also true for “community level education” where only 5% are in the high education group. (Table 1)

2.5. Table 2 could give more information if the second row is subdivided into each type of flesh foods. The first row gives the aggregate measure already.

3. Results and discussion

3.1. “The finding of this study revealed that children whose mothers had at least four ANC visit and gave birth at health facility had a protective effects against low ASF consumption. The possible pathways (mechanisms) of this association may be due to ease of access to health information that may improve the likelihood of mothers’ practice of proper child feeding” (line 415 – 418). Can you provide evidence to this in your research, please?

3.2. “The current analysis indicated that the ASF consumption of children in Ethiopia is 22.7%” – better if written as “Currently, only 23% of children in Ethiopia consume ASF” of something similar along this line otherwise, the conclusion drawn from this will be over riding.

3.3. Policy implication scattered here and there. They should be brought together under one heading or in one or two paragraphs. For example, line 381 to 388 against lines 442 to 445

6. PLOS authors have the option to publish the peer review history of their article (what does this mean?). If published, this will include your full peer review and any attached files.

Reviewer #1: No

Reviewer #2: No

---

## [Author Response · Author response to Decision Letter 0]

14 Jan 2022

From authors 

Manuscript title: Individual and community-level factors associated with animal source food consumption among children aged 6-23 months in Ethiopia: Multilevel mixed effects logistic regression model

Dear editor and reviewers

I am very grateful for all valuable comments and suggestions you gave us for the improvement of the paper. Hereunder, I responded to each points raised and these are corrected and amended in the revised manuscript when applicable (indicated in track changes) and clean copy of the final manuscript. 

Editor’s comments

1. The funding statements 

Response: since the authors did not receive any funding the funding statement was updated as “Funding: The authors received no specific funding for this work. The funders had no role in study design, data collection and analysis, decision to publish, or preparation of the manuscript.

2. Thank you for stating the following in your Competing Interests section

Response: We included an updated conflict of interest form and we filled the online system.

3. In your Data Availability statement,

Response: regarding the data availability, we updated the statement in accordance with the DHS data sharing policy (https://dhsprogram.com/data/terms-of-use.cfm) which stated that the data can only be used for the registered study only and will not be shared for other. Since it is a third party data which is obtained from the Demographic and Health survey repository and researcher can obtain the data through research project registration and can utilize the data up on agreeing for the terms and conditions for the data use.

Thus, we included the following statement as: ” The data used in this study are from the Ethiopian Demographic, and Health Survey (2016 and 2019) and can be requested from the DHS office at https://dhsprogram.com/Data/ using the details stated in the Materials and Methods section of this paper.”

4. We note that you have indicated that data from this study are available upon request. PLOS only allows data to be available upon request if there are legal or ethical restrictions on sharing data publicly. 

Response: since the data is from third party (DHS program), the terms and conditions do not allow to share the data to others (only for the registered data only). Please refer to the above updated data availability statement.

Reviewer 1 comments

1. General comments 

1.1. “Animal source food consumption is very low” (line 47) – reference? Compared to? 

Response: diversified diet is recommended in ideal scenario and studies reported ASF consumption above the one we obtain in this study (22%). This indicates that only one in 5 children had consume ASF (refers to egg and flesh products) which is by far below the ideal case, where we said ASF consumption is poor among children in the study area. 

1.2. “despite a better livestock population, diversified diet consumption is poor.” should it be diversified ASF? (line 76)

Response: It is just to refer to diversified diet containing diversified ASF consumption. It is corrected in the revised version.

1.3. Which time period are the data points referring? (Line 78)

Response: corrected (for the year 2019)

1.4. “So far, limited attempts were tried to understand the problem with the use of small-scale surveys, despite the multifaceted nature of the issue, where individual or household level factors only may not be sufficient” (line 86 – 88). What are these studies? A few citations of these studies will be ideal. How is your study unique from these studies? 

Response: done, relevant citations are included to show the gap. This part is extensively revised and a more recent publication in the area are included and cited. In addition, the gap and the need for evidence is clearly shown. 

1.5. “Nationwide study addressing the factors that affect the level of ASF consumption has not been conducted in Ethiopia” (line 97 – 98) I am afraid I can’t agree to this claim. For example, Workicho et.al (2016) Household dietary diversity and Animal Source Food consumption in Ethiopia: evidence from the 2011 Welfare Monitoring Survey, Potts et.al (2019) Animal Source Food Consumption in Young Children from Four Regions of Ethiopia: Association with Religion, Livelihood, and Participation in the Productive Safety Net Program are a few examples. You can cite these researchers and point out what makes your study unique. 

 Response: the comment was helpful to consider the studies done so far and justify our study. The revised justifications are included in the final paragraphs of the introduction. I am really grateful for that. As indicated in the revised manuscript, there are studies: both local studies and national wide studies, but the current data (updated and covers almost all regions of the country) and more robust statistical approaches allows to clearly identify the hierarchical predictors of ASF consumption among children.

1.6. “Currently, the price of ASF is rising alarmingly in the country above 36%”. Lin 381. This should be re-written. There is no single ASF price as such unless it is a price index, I think. 

Response: corrected, it refers to the percentage increase in the price of food items (price index).

1.7. “In Ethiopia, 35% [37], 57% and 36.8% [9] are a victims of Zinc deficiency, anemia and stunting…”line 438, who are these figures referring to? Of the Ethiopian population, Ethiopian children or ? 

Response: refers to children in Ethiopia based on the national survey done in 2019. It is corrected accordingly in the revised version.

2. Methodology 

Measurement of the dependent variable – while a dichotomized variable is a good starting point, it is also crude as it aggregates measurement and somehow undermines heterogeneity, for example in the type of ASF consumed here. In this regard, as a robustness extension, 

2.1. In addition, “A large scale regional variation (..) in Addis Ababa and … lowest in Somali was observed in ASF consumption aged 6-23 months.” (line 262 – 264). And “Low ASF consumption is more prevalent in more food insecure regions of the country (Afar, Somali, and other regions).” (Line 361 – 362). “Apart from the resource scarcity related factors, one of the reasons for this disparity could be measurement of ASF, as we do not consider milk and milk products widely used in pastoralist communities of Ethiopia”. 

This provides a wrong information and introduces systematic bias. Afar and Somali constitute one of the largest livestock populations making children more likely to consume milk and milk products. These regions do not have similar wealth of chicken and are less likely to consume egg. 

• Bring the definition of ASF used in this manuscript forward explicitly at the start (Mainly in the dependent variable section) and give a concerted direction to your reader what you mean by an ASF 

Response: due to the nature of the data and the WHO/FAO definition for ASF consumption, we tried thoroughly for the first analysis and we finally decided to consider the mentioned food groups (also stated in the FAO definition) to our case. Then, considering the limitation of the analysis in the discussion part. 

It shows us your critical review the paper and thank you very much for this. This question was also the main question for us at the start of this work. However, the FAO and WHO IYCF indicator states ASF consumption as the seventh important indicator which is operationalized there. The definition of ASF there states that” the percentage of children who consumed egg, poultry, animal fleshes during the past 24 hours). Dear reviewer, please kindly note that our target population are infants and children, where we expect almost all consume either breast milk and/or other milks during this time. Due to this, the indicator is defined not to consider milk and milk product in the indicator definition. 

More specifically, in our country the rate of continued breast feeding till two years and median duration of breast feeding (reaches up to 36 months) is high and long, where children are being fed at least with breast fed. That is why the ASF consumption is operationalized like this.

As per the reviewer’s recommendations we defined the ASF consumption clearly in the introduction and variable section.

2.2. Will a separate regression (as a dichotomous variable) be an extension of this study or be used as a robustness check? For example, estimating the same model for the consumption of Egg and other flesh foods alone. 

Response: The outcome is defined based egg and other flesh foods consumptions. So, for the description purpose, we presented the overall ASF consumption and each item separately. However, since our aim is to assess the predictors of ASF consumption, the outcome variable is ASF consumption (“1” for yes and “0” for No). So, there is no need to have a separate regression model.

2.3. Counting the number of ASF and generating an ordered variable to run multi-level ordered logistic regression provide similar results? 

Response: No, please note that the nature of the dependent variable is Yes and No type (dichotomous, not ranked), where this variable could not be ranked to OLS. Thus, we applied the multilevel binary logistic regression for dichotomous outcome variable instead of OLS, which is not appropriate. 

2.4. “Only 27 (0.6%) of mothers had accesses to all three media at least once a week” which three media? I think this needs re-phrasing the statement. Looking at Table 1, this variable does not show variability worth to have it in a description. 99% said No to media access. This requires re categorization or will not make a good predictor. This is also true for “community level education” where only 5% are in the high education group. (Table 1) 

Response: these statements are revised and some confusing statements are deleted. The community level education is a community level aggregated indicator for education status (which is based on the actual measurements). 

2.5. Table 2 could give more information if the second row is subdivided into each type of flesh foods. The first row gives the aggregate measure already. 

Response: Here, the overall ASF consumption (first row) and the individual food consumptions (each food are presented in subsequent rows). The data did not specifically indicate specific flesh foods as these are many.

3. Results and discussion 

3.1. “The finding of this study revealed that children whose mothers had at least four ANC visit and gave birth at health facility had a protective effect against low ASF consumption. The possible pathways (mechanisms) of this association may be due to ease of access to health information that may improve the likelihood of mothers’ practice of proper child feeding” (line 415 – 418). Can you provide evidence to this in your research, please?

Response: Evidence is provided in the revised manuscript.

3.2. “The current analysis indicated that the ASF consumption of children in Ethiopia is 22.7%” – better if written as “Currently, only 23% of children in Ethiopia consume ASF” of something similar along this line otherwise, the conclusion drawn from this will be over riding.

Response: it is corrected and revised accordingly.

3.3. Policy implication scattered here and there. They should be brought together under one heading or in one or two paragraphs. For example, line 381 to 388 against lines 442 to 445 

Response: the policy implications were expressed and indicated in the discussion within each section. However, that may not be good to present like that. So that we prefer to present the policy implications and recommended strategies to be under each study finding, where it allows to clearly indicate the implications there than putting together.

Reviewer 2 comments and responses

Line 23: The background section could be strengthened by a) emphasizing the importance of ASF (rather than diversified diet), b) emphasizing the age group. 

Response: ok, it is corrected

Line 27: Suggest changing ‘multi-layered’ to more specific – perhaps individual and community level. 

Response: it is corrected as individual and community level …..

Line 30: Sentence beginning with “a stratified two-stage” is unclear. What is complex sample design referring to? How were weights used? 

Response: corrected and the survey weight calculated by the DHS survey weighting method was applied. The survey weight for complex sample design is used ad applied for descriptive statistics.

Line 38: I’m curious about the inclusion of home delivered children as a variable, as it may have substantial overlap with some of the other predictors, particularly low SES. In addition, what overlap is there between low SES and high community poverty level, outside using it as both an individual and community level factor? 

Response: yes, we are aware of the potential overlap and an interaction analysis was done for possible effect modification and we did not find a statistically significant effect modification in the model. The analysis was done for SES, Home delivery and other potential variables (P-value above 0.05). We are grateful for the importance notice and suggestions.

Line 47: I would love to see the conclusions be a bit more specific in terms of what targeting and policy decisions could be incorporated based on the findings. 

Response: revised

Line 62: Suggest re-arranging the background to begin with animal-source foods role in optimal development, rather than the indicator, which belongs later or in the methods section. 

Response: revised as per the reviewer request.

Line 71: There are also several more recent publications and meta-analyses which discuss the importance of ASF – inclusion of those would strengthen the background section, which relies on several single-sited analyses or position papers. Please see https://academic.oup.com/advances/article/10/5/827/5513046?login=true and https://www.cochranelibrary.com/cdsr/doi/10.1002/14651858.CD012818.pub2/full. 

Response: we included this recent publication in the revised versions.

Line 76: Suggest “widespread” livestock production, or something similar 

Response: it is revised and corrected.

Line 78: Suggest revising the line. 

Response: Ambiguous sentence was deleted and restructured.

Line 85: Curious if ASF should be considered a practice? Practice to me would suggest a cultural component, but I believe what this analysis is getting at is its availability, affordability, accessibility, etc. Does not necessitate a revision if practice is what is intended – just some elaboration. 

Response: here, we refer to ASF consumption to indicate the actual ASF consumption by children obtained from mother or caregivers’ response. So, it is actual consumption history and is not mainly concentrated on the availability and accessibility (which are secondary). 

Line 93: Language is a bit informal

Response: corrected and revised.

Line 98: It would be helpful to discuss why high-risk fertility behaviors is specifically highlighted here. 

Response: High-risky fertility behaviors are mainly related to a larger family, low socioeconomic class and other collinear variables. Thus, having high fertility-risky behaviors is related to lower availability and accessibility to ASFs, which limit its consumptions. Descriptions and evidences on it is included there.

Line 105: No need to include the names of variables in this section. 

Response: deleted.

Line 120: It would be helpful to explain why the study used children ages 6-23 months in particular. This may be better included in the background where the importance of ASF in this age range is emphasized – otherwise there is an equally strong argument to use a wider age range. 

Response: As we know failure to get adequate nutrition during the first one thousand days (1000 days) of life very critical period, where majority of stunting and micronutrition with all its adverse short and long-term problems can be prevented. This period includes pregnancy through the first two years of childhood. However, the period before six months infants and fetus are primarily dependent on maternal nutrient supply through breast feeding and placenta, where the role of extra meals starts after six months. This is clearly stated in the background part (it is stated in page 4 paragraph 2).

Line 127-130: I’m curious why Group 4 dairy products isn’t considered here, as this is also an animal source food. This would necessitate re-running all analyses, however unless there is a strong theoretical reason for excluding it, it should also be included. 

Response: Response: due to the nature of the data and the WHO/FAO definition for ASF consumption, we tried thoroughly for the first analysis and we finally decided to consider the mentioned food groups (also stated in the FAO definition) to our case. Then, considering the limitation of the analysis in the discussion part. 

It shows us your critical review the paper and thank you very much for this. This question was also the main question for us at the start of this work. However, the FAO and WHO IYCF indicator states ASF consumption as the seventh important indicator which is operationalized there. The definition of ASF there states that” the percentage of children who consumed egg, poultry, animal fleshes during the past 24 hours). Dear reviewer, please kindly note that our target population are infants and children, where we expect almost all consume either breast milk and/or other milks during this time. Due to this, the indicator is defined not to consider milk and milk product in the indicator definition. 

More specifically, in our country the rate of continued breast feeding till two years and median duration of breast feeding (reaches up to 36 months) is high and long, where children are being fed at least with breast fed. That is why the ASF consumption is operationalized like this.

As per the reviewer’s recommendations we defined the ASF consumption clearly in the introduction and variable section.

Line 130: It’s important to note that this indicator does not exactly refer to adequate ASF consumption, but that ASF consumption is necessary for minimum dietary diversity. It’s therefore inaccurate to say “adequate animal source food consumption,” as this is only a criterion for MDD. I would simply rephrase to ASF consumption or no ASF consumption, but avoid labeling the variable as adequate, since there is technically no indicator for this. 

Response: yes, I noticed the problems with the terms used in this analysis. Thus, we reframed as “consumed ASF” or “not consumed ASF” respectively. Thanks for that.

Line 135: It would be helpful to include citations that support the use of ‘high risk fertility behaviors’ from a theoretical perspective. I can understand why mother’s age <18 might influence ASF consumption as a proxy for poverty or decision-making, but not mother’s age >34 or birth order above three. These are more adequately capturing risks to the mother and child during birth and early life but I don’t see the connection to ASF consumption. 

Response: As per the comment there are evidences that show high fertility risk behaviors is prevalent (72%) and linked to anemia and child nutritional status [1]. The link between high fertility risk behavior with ASF consumption is mainly due to the household size, poor socioeconomic status and low access to ASF and hence ultimately low consumption. 

Line 148: Don’t need to include Stata commands or variable names throughout this or later sections. I’m curious here about the reasons for using community poverty level and community education level, as I don’t think they’re well explained. You might also note here that 

Response: The stata commands are removed from the result, it is acceptable. The community level factors are very critical and important factors associated with the ASF consumption. It is known that the individual level factors only will not predict the ASF consumption, which may also be affected by the community level general context (geographical related factors), which needs to be evaluated. That is why we considered the community level factors, despite the issue of some correlations. Also, these community level indicators are a statistically robust indicator. 

Line 250: I’ve discussed inclusion of different variables in several other sections but also want to highlight a few more that seem missing. In particular, I believe DHS asks whether households own any livestock, which would be important to include. I apologize if I am wrong on that. 

Response: Here, your concern is good and previous studies showed that livestock ownership is associated with a better ASF consumption. However, on the other way round this may not hold true for the majority of the poor where they consider such foods sources as cash crop instead of own consumption. Due to such scenarios, and the DHS did not explicitly report that, we did not consider livestock ownership.

Line 273: I’m curious why variables with a p-value of <0.25 were included – you may want to include a citation for this. Otherwise, I would suggest revising many of these analyses so that there is a strong theoretical basis for their inclusion. Since the paper is interested in exploring variables that are associated with ASF and designing policy around that, it should be made clear why, for example, ANC visits or place of delivery are useful predictors of ASF consumption. The same holds true of fertility risk. 

Response: lower p-value (p-value below 0.25) and/or variables with strong theoretical relation with ASF consumption and previously identified predictor variables were considered for the multivariable model, which is supported by many statistical approaches. Thus, we did not only rely on the p-value. Thus, we use a combination variable selection procedure including stepwise backward regression model for model building in combination with p-value and theoretical backgrounds.

Line 280-284: I would like to see in an appendix any additional model fitting that you carried out with different variables. As stated above, many of these variables don’t seem to have a strong rationale for inclusion. Model fit generally increases with more variables, but it would be important to demonstrate by how much, precisely, to justify their inclusion. You note that the very end that associations with a p-value below 5% were included, but in what stage of the process did you fit the variables? 

Response: Here using the detail statistical modeling approaches stated above, we fitted a bivariable and multivariable model, where the bivariable is more likely to be confounded by multiple variables and that is why we prefer to present the multivariable fitted logistic regression model. We did not search for p-value 0.05 only rather we statistically fit the model and present the findings. In addition, the model fitting information for different iterations (AIC and DIC) are presented in the result part in detail.

Line 284: Correct to p-value below 0.05. 

Response: It is corrected as p-value below 0.05.

Line 328: It would be helpful to re-organize this table to separate individual and community level factors more clearly. 

Response: This table is already organized in to individual level factors first followed by community level factor thereafter. We prefer to present like this to avoid redundancy.

Line 365: This statistic comes from one survey in Indonesia, which should either be noted, or the statistic removed. 

Response: done

Line 358-370: This discussion would be better used to present more of the overall findings from your analysis – rather than other analyses. This discussion only mentions that ASF consumption is low and more prevalent in food insecure regions. It should be connected back to the main themes from your findings. 

Response: it is revised as per the comments

Line 416: This is a great explanation of possible pathways for ANC visits, though I would be curious whether these variables significantly improve model fit individual when included with education and wealth. If so, this is a great finding, but I would also like to see it explained more – e.g. what messages do mothers receive? Finally, is there a reason for including both ANC visits and home birth? 

Response: The inclusion of these variables significantly improved the model fitness and are included in the final multivariable model. It is known that the ANC period is a potential contact point where maternal and child nutrition information are being delivered under the ENA program. And studies are showing that mothers are aware of the information and tends to change their behaviors.

Reference 

1. Tamirat, K.S., G.A. Tesema and Z.T. Tessema, Determinants of maternal high-risk fertility behaviors and its correlation with child stunting and anemia in the East Africa region: A pooled analysis of nine East African countries. PLOS ONE, 2021. 16(6): p. e0253736.

---

## [Decision Letter · Decision Letter 1]

1 Mar 2022

PONE-D-21-25620R1Individual and community-level factors associated with animal source food consumption among children aged 6-23 months in Ethiopia: Multilevel mixed effects logistic regression modelPLOS ONE

Dear Dr. Abdu,

Thank you for submitting your manuscript to PLOS ONE. After careful consideration, we feel that it has merit but does not fully meet PLOS ONE’s publication criteria as it currently stands. Therefore, we invite you to submit a revised version of the manuscript that addresses the points raised during the review process.

We look forward to receiving your revised manuscript.

Kind regards,

Sukumar Vellakkal

Academic Editor

PLOS ONE

Journal Requirements:

Reviewers' comments:

Reviewer's Responses to Questions

**Comments to the Author**

1. If the authors have adequately addressed your comments raised in a previous round of review and you feel that this manuscript is now acceptable for publication, you may indicate that here to bypass the “Comments to the Author” section, enter your conflict of interest statement in the “Confidential to Editor” section, and submit your "Accept" recommendation.

Reviewer #1: All comments have been addressed

Reviewer #2: All comments have been addressed

2. Is the manuscript technically sound, and do the data support the conclusions?

Reviewer #1: Partly

Reviewer #2: Yes

3. Has the statistical analysis been performed appropriately and rigorously? 

Reviewer #1: I Don't Know

Reviewer #2: Yes

4. Have the authors made all data underlying the findings in their manuscript fully available?

Reviewer #1: No

Reviewer #2: Yes

5. Is the manuscript presented in an intelligible fashion and written in standard English?

Reviewer #1: No

Reviewer #2: Yes

6. Review Comments to the Author

Reviewer #1: Thank you for your thoughtful comments. I think this manuscript is on track to acceptance, but I have several suggestions for revisions that remain.

1. The introduction section, particularly lines 81-88, still relies too heavily on single studies. There are more recent systematic reviews which explicitly look at children age 6-23 months (https://www.cambridge.org/core/journals/british-journal-of-nutrition/article/abs/animalsource-foods-as-a-suitable-complementary-food-for-improved-physical-growth-in-6-to-24monthold-children-in-low-and-middleincome-countries-a-systematic-review-and-metaanalysis-of-randomised-controlled-trials/6427FFE371BAAC054742E8EBE8147B1D). These should be included, and meta-analyses emphasized over case studies or older, single sited papers.

2. Table 3 should be reworded to describe associations with no animal source food consumption, otherwise the OR read as reversed.

3. I understand your point about milk being different from ASF, however the definition that you cited is incorrect. The new IYCF indicators mention minimum milk feeding frequency for non-breastfed infants AND egg and/or flesh food consumption. Thus, minimum milk feeding is included as a separate variable, and since you do not include EBF or any breastfeeding in your model, this risks eliding the evidence. If you continue to choose not to include dairy consumption, I would re-specifiy, throughout the paper, you refer to egg and or flesh food consumption. I don't think it's sufficient to operationalize ASF as not including milk - the vast majority of literature, including much that you have cited (e.g. Dror, Eaton, etc.) include dairy in ASF consumption. Thus the evidence you are using for support doesn't align with the analyses you performed.

4. I continue to disagree with the inclusion of high risk fertility behaviors. The paper you cited in support shows somewhat ambiguous evidence which is very dependent on WHICH high risk fertility behaviors are included. As they note, Rich women, on the other hand, are more likely than poor women to participate in highrisk fertility activity" - and that model shows some negative associations between stunting and high risk fertility behaviors (e.g. age >34). It seems you've highlighted two very specific situations as well as one broad category (briths with any multiple risk category), but this neglects important nuance that isn't theoretically supported. I strongly suggest removing this variable, as I suspect it is leading to model overfitting, rather than shedding a light on the associations with specific behaviors. Moreover, it is difficult to understand why this information would be important to a policy maker, given that it's likely a proxy for the real variables which influence ASF consumption.

5. Finally, I suggest enlisting the services of a copyeditor for a full proofread.

Reviewer #2: I have have read their resubmitted version in depth. I believe they have addressed most of the comments I gave last time or provided compelling arguments for the ones they wanted to keep. Overall I am satisfied with the revision the author have made.

7. PLOS authors have the option to publish the peer review history of their article (what does this mean?). If published, this will include your full peer review and any attached files.

Reviewer #1: No

Reviewer #2: No

---

## [Author Response · Author response to Decision Letter 1]

7 Mar 2022

From authors 

Manuscript title: Individual and community-level factors associated with animal source food consumption among children aged 6-23 months in Ethiopia: Multilevel mixed effects logistic regression model

Manuscript ID: PONE-D-21-25620R1

Dear editor and reviewers

I am very grateful for all valuable comments and suggestions you gave us for the improvement of the paper. Hereunder, I responded to each point raised in the second-round review and these amendments are indicated in track change in the revised manuscript when applicable and clean copy of the final manuscript is submitted. 

Editor’s comments

#1 The issue regarding depositing the laboratory protocol in a relevant repository, since all methodological approaches are stated and indicated in the manuscript, it is not applicable for this case.

It is not applicable

#2 We did not make any change to financial disclosure

# 3 the issue of retracted references

Here we used the scite tool to check for retracted references and w updated the reference section accordingly. We also manually checked for retracted references and citation with retracted referencing were managed accordingly. 

Reviewers’ comments

Fortunately, almost all the comments have been addressed in the first revision and editor and the reviewers duly acknowledged our effort, thanks for that. But there are some comments that may need clarification from the authors. First of all, we are glad to have your valuable comments.

Reviewer 1 comments

#1 on the introduction

Response; we accept the comments and we included other robust and more comprehensive evidence on the issue. The references are updated in the revised version.

#2 Reword table 3

Response: It is revised as requested “no ASF consumption”.

#3 the issue of Milk is different from ASF 

Response: in the previous revision, we clearly indicated the definition of the indicators and ASF is defined in WHO in a way we define it in the current paper. But I accept, the issue to include correct references to the indictor definition. In order to be in line with the WHO updated new indicators, we still stand to define ASF consumption (as indicator)-as egg and/or flesh consumption [1] as such food groups are different from milk consumption among children due to

#1 majority of children consume breast milk up to two years

# 2 proportion of older children who consumer milk (cow, goat, camel) is high but,

#3 the typical consumption of egg or flesh meats has a paramount importance where access to such a food is low in developing countries due to rising price of animal source foods. Moreover, such foods are rich source of iron, zinc, vitamins and other mineral necessary for optimal growth in addition to protein, carbohydrates and fats which are common in milk. But the first three nutrients are found in scant amount from milk and milk products.

It is due to the above evidences that WHO excludes milk and milk product from the definition. I hope you will understand that.

#4 high risky fertility behaviors

Response: above all the point raised by the reviewer here is strong and logical to us to reconsider the decision regarding variable selection. The reason we included this variable in the model is due to the following.

A. The inclusion of this variable greatly improved the model fitness and it is found in many different literatures that High fertility behavior is strongly linked to food security, access to nutritious food and nutritional status of children (stunting). 

We can clearly understand from the pooled analysis of DHS surveys from 2006 to 2012 that fertility behavior is important factor that determine access to food and nutritional status of children [2].

B. as this indicator shows the overall fertility risk of mother of children, this may indirectly indicate or correlate with birth interval and number of children, which in turn affects the ASF consumption negatively, we included it in the description part. 

#5 copy edition and proofreads

The revised manuscript undergoes a thorough copy edition and proof read by professional editor as indicated in the revised manuscript (when applicable) and cleaned manuscript.

Reviewer 2 comments

-the reviewer is convinced by the revisions made and the compelling rational arguments raised by the author. Thanks for understanding the approaches and the methods used for this paper.

Additional comments; we uploaded the figures for PACE tool and the output was ok.

References

[1] Organization, W.H., Indicators for assessing infant and young child feeding practices: definitions and measurement methods. 2021.

[2] Rutstein, S.O. and R. Winter, The effects of fertility behavior on child survival and child nutritional status: evidence from the demographic and health surveys, 2006 to 20122014: ICF International.

---

## [Editor Report · Decision Letter 2]

10 Mar 2022

Individual and community-level factors associated with animal source food consumption among children aged 6-23 months in Ethiopia: Multilevel mixed effects logistic regression model

PONE-D-21-25620R2

Dear Dr. Abdu,

We’re pleased to inform you that your manuscript has been judged scientifically suitable for publication and will be formally accepted for publication once it meets all outstanding technical requirements.

Kind regards,

Sukumar Vellakkal

Academic Editor

PLOS ONE
---

## [Editor Report · Acceptance letter]

23 Mar 2022

PONE-D-21-25620R2 

Individual and community-level factors associated with animal source food consumption among children aged 6-23 months in Ethiopia: Multilevel mixed effects logistic regression model 

Dear Dr. Abdu:

I'm pleased to inform you that your manuscript has been deemed suitable for publication in PLOS ONE. Congratulations! Your manuscript is now with our production department. 

Kind regards, 

on behalf of

Dr. Sukumar Vellakkal 

Academic Editor

PLOS ONE